# Multicolor recordable and erasable photonic crystals based on on-off thermoswitchable mechanochromism toward inkless rewritable paper

Yang Hu[1], Chenze Qi[1], Dekun Ma[1], Dongpeng Yang [1] ✉ & Shaoming Huang [2] ✉

Mechanochromic photonic crystals are attractive due to their force-dependent structural colors; however, showing unrecordable color and unsatisfied performances, which significantly limits their development and expansion toward advanced applications. Here, a thermal-responsive mechanochromic photonic crystal with a multicolor recordability-erasability was fabricated by combining non-close-packing mechanochromic photonic crystals and phase-change materials. Multicolor recordability is realized by pressing thermal-responsive mechanochromic photonic crystals to obtain target colors over the phase-change temperature followed by fixing the target colors and deformed configuration at room temperature. The stable recorded color can be erased and reconfigured by simply heating and similar color-recording procedures respectively due to the thermoswitchable on-off mechanochromism of thermal-responsive mechanochromic photonic crystals along with solid-gel phase transition. These thermal-responsive mechanochromic photonic crystals are ideal rewritable papers for ink-freely achieving multicolor patterns with high resolution, difficult for conventional photonic papers. This work offers a perspective for designing color-recordable/erasable and other stimulus-switchable materials with advanced applications.

Responsive photonic crystals (PCs) that can change their optical performances under diverse stimuli have attracted extensive attention due to their wide potential applications in displays[1-4], sensing[5-10], printing[11-15], anti-counterfeiting[16-18], and optical devices[19-21]. Among them, mechanochromic photonic crystals (MPCs) have attracted particular interest because they possess non-close-packing structures and hence can change their structural colors in response to forces, similar to the working mechanism of chameleon skin[22-24]. Moreover, forces possess the merits of being easily available, low-power consumption, and applicable in large areas with high resolution. So far, various MPCs have been prepared based on self-assembly, shear-induced assembly,

two-step filling, and swelling close-packing or non-close-packing structures[25-27]. These works mostly focus on improving the sensitivity and tuning range of structural colors of MPCs, while little attention was paid to expanding their advanced applications due to the lack of structure and composition design. Several key challenges must be addressed to promote the rapid practical applications of MPCs. First, multicolor recordability, that is recording the changed colors of an MPC, is strongly desired to fulfill the increasing low-power demand for colors. However, nearly all state-of-the-art MPCs need external forces to retain the changed structural colors which will return to the original ones once the load is removed. This is majorly because the deformed

[1]Zhejiang Key Laboratory of Alternative Technologies for Fine Chemicals Process, School of Chemistry and Chemical Engineering, Shaoxing University, Shaoxing 312000, China. [2]School of Chemistry and Materials Science, Hangzhou Institute for Advanced Study, University of Chinese Academy of Sciences, Hangzhou 310024, China. ✉e-mail: dpyang@usx.edu.cn; smhuang@gdut.edu.cn

MPC configuration cannot be fixed. Second, erasing recorded colors by easily operational stimuli without environmental pollution is needed to realize good recyclability and green chemical industry. In addition, the color recordability and erasability should be repeatable. Third, multicolor recording and fast color switching speed are required to achieve high efficiency and good tunability. Fourth, excellent stability of MPCs themselves under normal conditions, color recording, and color releasing states is necessary to meet the needs of the practical application. The combination of all these intriguing features will lead to high-performance, multicolor-recordable, fast-responsive, and long-term stable MPCs toward advanced applications.

To date, considerable methods and efforts have been devoted to realizing multicolor recordability on MPCs with inverse opal structures. Jiang et al. have prepared vapor-responsive shape-memory polymers (SMPs)-based MPCs that enable unusual cold-programming and instantaneous shape recovery at the nanoscale[28–31]. The MPC can switch from brilliant color to a recordable colorless state under pressure owing to the collapse of macropores, leading to the recording of monotonous colors. Although the monochrome color recordability and the use of organic vapors will limit their practical applications, these works lay an important foundation for exploring color recordability. Wu and co-workers reported color recordability of monochrome retroreflective color using an SMP-based micrometer-sized dome monolayer array[32]. Unfortunately, the unsatisfied color purity and on-off colors may limit their practical applications. Wang's group prepared mechanochromic and thermochromic PCs with close-packing structures based on the SMP-based core-shell particles[33]. Nevertheless, their applications might be difficult because of the unsatisfied color tunability and the significant decrease in color saturation. Du's group developed a colorless SMPs-based MPC with compression-induced collapsed micropores and a heat-programming shape recovery degree[34]. Multicolor can be generated and recorded by region-selectively heating the MPCs to different temperatures ($T$) using lasers and integrating a pre-printed photothermal layer. However, the unwanted diffusion colors caused by the inevitable thermo-delivery at boundaries between heated and unheated regions will restrict the color purity. Yang and co-workers realize multicolor recording based on the elastoplastic deformation of the uncrosslinked photoresist film with inverse opal structures[35]. Nevertheless, the recorded colors cannot be erased due to the permanent change of the structure, unfavorable for green and economic materials. Therefore, it remains a big challenge for reported MPCs to simultaneously possess multicolor recordability, erasability, and good color purity.

Here, we report a thermal-responsive MPC (TRMPC, Fig. 1) showing untraditional multicolor recordability, erasability, and high color purity, prepared by the combination of the specific phase-change material (PCM) of polyoxyethylene (40) nonylphenyl ether (PNE) and MPCs with both swellable non-closely packed structures and small refractive index contrast ($\Delta n$, Eq. 1, Supplementary Note 1) between silica particles and surroundings. The uniform introduction of PCM into MPCs' lattice leads to an on-off themoswitchable mechnanochromism and a small influence on the color output, which is the key to realizing the above characteristics of TRMPCs. Different from the reported strategies, color recordability is accomplished in two processes: color generation by pressing the TRMPC at $T$ higher than the melting point ($T_m$) of the PCM and subsequently color fixation through a cooling down process. Multicolor can be recorded by adjusting pressure with similar procedures. Interestingly, recorded colors exhibit excellent stability (>1 year) under normal conditions but can be easily erased in 2 s by simply heating the TRMPC to a $T$ higher than $T_m$ to recover the TRMPC's configuration. Compared to other stimuli, the use of forces and $T$ for recording and erasing colors is efficient and convenient, which will facilitate their practical applications. We demonstrate an inkless way to construct multicolor, high resolution, reconfigurable patterns by alternate region-selectively pressing the TRMPC with hot stamps to obtain patterns and erasing patterns through heating. The ink-free writable TRMPC paper shows advantages in reducing the usage of conventional disposable papers and inks and environmental pollution. This work offers a perspective for designing color-recordable and erasable MPCs and other stimulus-switchable materials and will promote their applications in green printing, smart sensing, and low-power displays.

## Results

### Fabrication and characterizations of TRMPCs

TRMPCs (Fig. 2a) with thermoswitchable on-off mechanochromism were fabricated based on swelling MPC with non-closely packed structures and appropriately small $\Delta n$ by the PCM with a high $T_m$. Briefly, silica particles (167 nm, Supplementary Fig. 1) were dispersed in ethanol, di(ethylene glycol) ethyl ether acrylate (DEGEEA), and polyethylene glycol mono-phenyl ester acrylate (PEGPEA) by sonication. After evaporating ethanol at 100 °C for 1 h, a liquid silica/DEGEEA/

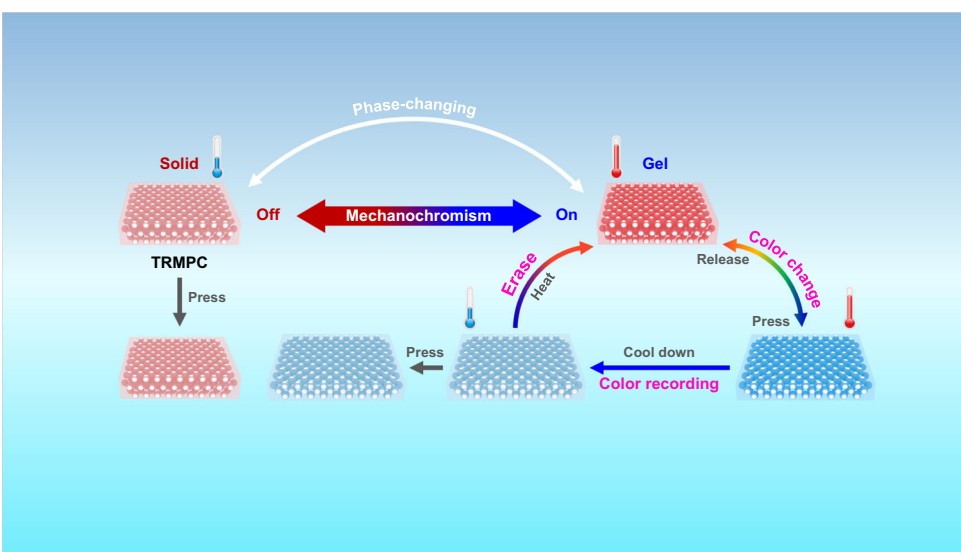

**Fig. 1 | Schematic illustration of the working mechanism of the TRMPC.** TRMPC transforms from solid to gel when $T > T_m$ and its color changes (such as red to blue) when pressure is applied. After cooling to $T < T_m$, the blue color remains unchanged when the pressure is removed (color recordability), and finally returns to red after being heated to $T > T_m$.

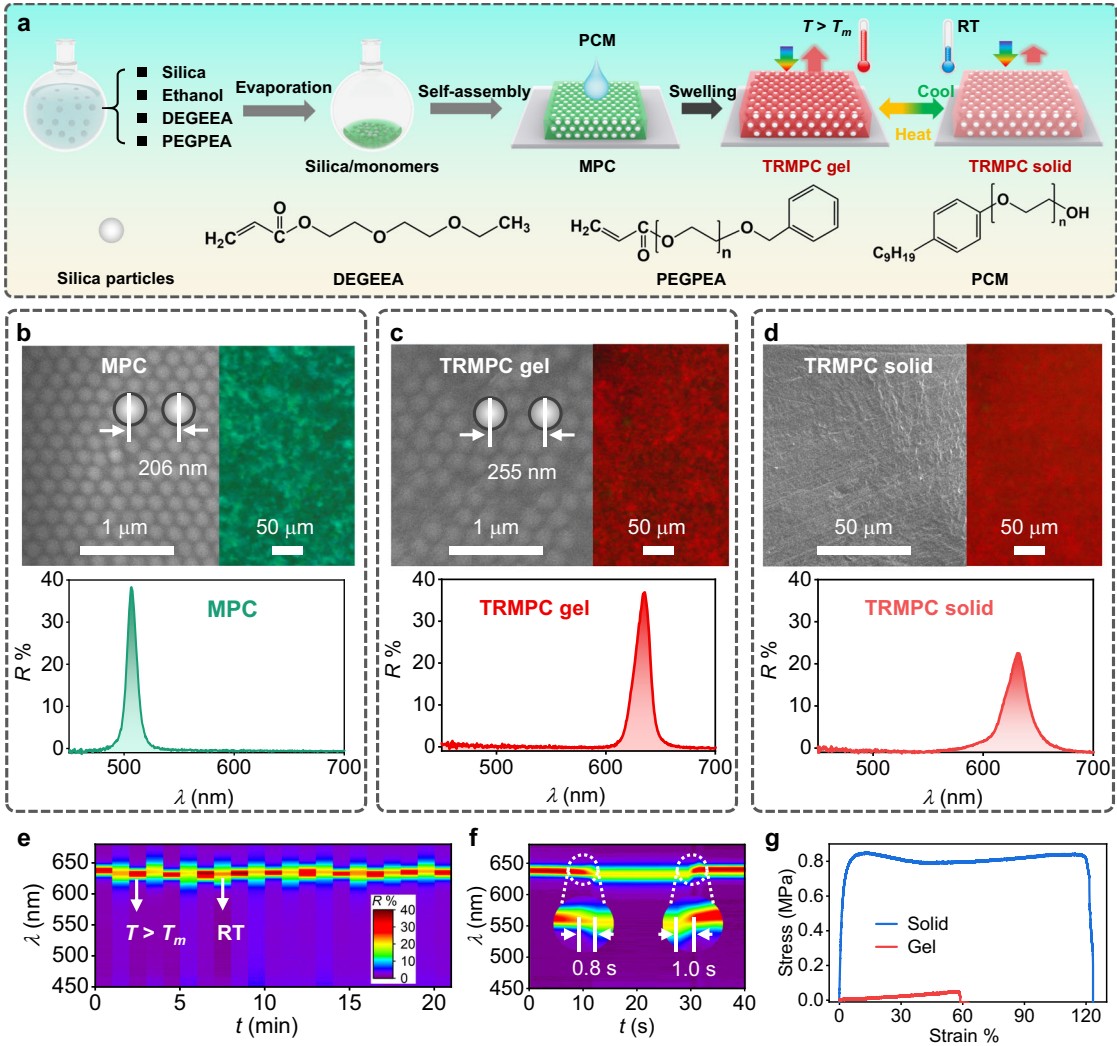

**Fig. 2 | Fabrication and characterization of the TRMPC. a** Schematic illustration of the fabrication of the TRMPC. **b**–**d** The SEM images, microscope images, and reflection spectra of the **b** MPC, **c** TRMPC gel, and **d** TRMPC solid, respectively. **e**, **f** The reflective signal switches of the TRMPC under alternative heating and cooling. **g** The stress-strain curves of the TRMPC solid and gel.

PEGPEA PC showing iridescent colors was obtained, suggesting the ordered packing of silica particles in the mixture of DEGEEA and PEGPEA. This liquid PC was converted into solid MPC through UV-triggered polymerization. The volume fraction of silica particles ($\varphi_s$: 15–40%) is designed much smaller than that (74%) of PC with a closely packed structure to construct a non-close-packing structure. The high content of elastic DEGEEA and PEGPEA combined with the non-closely packed structure enables the mechanochromic properties. For easy discussion, we use $f_d$ and $f_p$ to represent the volume fraction of DEGEEA ($\varphi_d$) and PEGPEA ($\varphi_p$) relative to the PEGPEA/DEGEEA mixture, respectively. Before optimizing the $f_d$ and $f_p$, the effect of $\Delta n$ ($f_p$) on reflectivity was investigated without the effect of $f_p$ ($\Delta n$). Correspondingly, the results demonstrated that reflectance is proportional to $\Delta n$ but inversely proportional to $f_p$ (Supplementary Note 2). A large $\Delta n$ and small $f_p$ will result in a high reflectance.

Then, MPCs ($\varphi_s$: 15%, Supplementary Fig. 2) with different $f_p$ were investigated to determine the optimized value. When $f_p = 0$, the reflectance of MPC is only 19% because of the ultrasmall $\Delta n$ (0.010) between silica particles and DEGEEA. With $f_p$ increases from 0 to 0.3, $\Delta n$ increases from 0.010 to 0.025 (Supplementary Fig. 3), leading to an increase in reflectance from 19% to 47%. However, when $f_p$ further increases (>0.3), the reflectance decreases dramatically despite the

increase in $\Delta n$, which can be attributed to the break of long-range order by the high $f_p$.

In the aspect of colloidal assembly in acrylates, there is the lowest value ($\varphi_{th}$) of $\varphi_s$, over which silica particles can self-assemble into uniform long-range ordered structures. MPC shows high and low order degree and thus reflectance when $\varphi_{th}$ is lower and higher than $\varphi_s$, respectively. The $\varphi_{th}$ of DEGEEA ($\varphi_{th-d}$) and PEGPEA ($\varphi_{th-p}$) is 10% and 25%, and the $\varphi_{th}$ of the PEGPEA/DEGEEA mixture ($\varphi_{th-mix}$) can be calculated by Eqs. 2–3. Supplementary Fig. 4 shows that the $\varphi_{th-mix}$ increases as $f_p$ increases and $\varphi_{th-mix} = 15\%$ when $f_p = 0.345$. These results mean high reflectance can be obtained when $f_p < 0.345$ owing to $\varphi_{th-mix} < \varphi_s$ (15%). In striking contrast, the reflectance decreases significantly when $f_p > 0.345$ due to $\varphi_{th-mix} > \varphi_s$ (15%). Thus, $f_p = 0.3$ should be the optimized value for MPC possessing both high order degree and appropriate $\Delta n$.

Under the scanning electron microscope (SEM, Fig. 2b), the MPC ($\varphi_s$: 30%) shows long-range ordered and non-close-packing structures. The interparticle distance ($D_{id}$) is measured to be 207 nm. The surface-to-surface distance ($D_{s-s}$) between neighboring particles is 39 nm, exceeding the effective distance of van der Waals attraction. These results indicate the strong electrostatic repulsion between silica particles is the major driving force for assembling silica particles into the

non-closely packed structure[36]. The $\zeta$-potential of silica particles in the DEGEEA/PEGPEA solution is −43 mV, giving evidence of the above hypothesis. By introducing a trace amount of the ionic liquid (1-ethyl-3-methylimidazolium bis(trifluoromethylsulfonyl)imide) into the liquid silica/DEGEEA/PEGPEA PC solution to selectively screen the electrostatic force, the structural color disappears and only amorphous structures were obtained (Supplementary Fig. 5), demonstrating the critical role of electrostatic force for ordered structure. The highly ordered structure of the MPC can efficiently and selectively diffract incident light, resulting in a bright green MPC with an intense reflectance at 506 nm. The reflection wavelength ($\lambda$) can be calculated by Bragg's law (Eq. 4), where $m$ is the diffraction order. $\theta$ is the angle between the reflected beam and the normal. $n_i$ and $\varphi_i$ are the refractive index and volume fraction of each material (silica particles and acrylates) of the PC, respectively. The $\lambda$ is calculated to be 502 nm, in line with the value from reflection spectra. The MPC exhibits a good monochromaticity as demonstrated by the ultranarrow full width at half-maximum (FWHM, 10.0 nm), much smaller than those (30–70 nm) of conventional MPCs[37–39]. We attribute this to the small $\Delta n$ (0.025, calculated by Eq. 1) between silica particles and polymers, which significantly suppresses incoherent scattering of light.

The non-close-packing structure enables us to adjust MPCs' $\lambda$ and colors covering most visible ranges by simply altering $\varphi_s$ with the same particle size, more convenient than the conventional way of regulating particle sizes. As presented in Supplementary Fig. 6, $\lambda$ decreases from 622 to 476 nm and the corresponding color varies from red to blue when $\varphi_s$ increases from 15 to 40%. The increase in $\varphi_s$ will cause a decrease in $D_{id}$ and thus $\lambda$ according to Bragg's law. These MPCs show high reflectance, brilliant colors, and small FWHM (7.2–12.2 nm), demonstrating their remarkable color purity. However, further decreasing $\varphi_s$ (5% and 10%) lead to negligible reflectance due to the lack of long-range order (Supplementary Fig. 7). A small $\varphi_s$ will lead to a too-large $D_{id}$, which causes a dramatic decrease in electrostatic repulsion, giving rise to the disordered structures and thus neglectable reflectance. Therefore, $\varphi_s$ of 15–40% should be a good choice for fabricating MPC.

$$\Delta n = n_{\text{silica}} - n_{\text{surroundings}} \qquad (1)$$

$$\varphi_{\text{th−mix}} = \varphi_{\text{th−d}} f_d + \varphi_{\text{th−p}} f_p \qquad (2)$$

$$f_d + f_p = 1 \qquad (3)$$

$$m\lambda = 1.633 D_{id} \left( \sum n_i^2 \varphi_i - \sin^2\theta \right)^{1/2} \qquad (4)$$

TRMPCs were fabricated by swelling the MPC in the PCM at a $T > T_m$. Here, MPC with $\varphi_s$ of 30% and a $\lambda$ of 506 nm is used for swelling. When soaked in the liquid PCM preheated to 80 °C, the PCM swelled into the polymer network of the MPC, leading to the increase in $D_{id}$ (Fig. 2c) and thus $\lambda$ over time (Supplementary Fig. 8). After reaching the swelling balance, the $\lambda$ remains constant, resulting in a red TRMPC organic gel with a $\lambda$ located at 632 nm. The ratio of $\lambda$ between TRMPC and MPC is 1.25, comparable to the ratio (1.22, Supplementary Fig. 9) of the corresponding thicknesses, firmly proving that the increase in $D_{id}$ is the key to the $\lambda$ redshift. The $\varphi_s$, $\varphi_p$, and $\varphi_{\text{PCM}}$ (PCM's volume fraction) of the TRMPC are 13.2%, 30.9%, and 55.9%, respectively. The FWHM of the TRMPC gel is 15.9 nm, slighter larger than that of MPC due to the slight increase in $\Delta n$ (0.038, Supplementary Note 1) caused by the PCM. After cooling down, the TRMPC undergoes a phase transition from gel to solid state with negligible change on $\lambda$, while the reflectance is decreased by 35%. Despite the slight increase in $\Delta n$, the TRMPC exhibits no obvious enhancement in incoherent light scattering compared to the MPC (Supplementary Fig. 10). This can be

explained by that the TRMPC's $\Delta n$ (0.038) is still small, which can efficiently suppress the incoherent scattering of light. Supplementary Fig. 11 shows that MPC and TRMPC show highly ordered structures without significant differences in order degree, demonstrating that the order degree is well retained with negligible influence by the PCM.

Under SEM (Fig. 2d), a coarse layer of PCM was observed on the surface of the TRMPC solid, slightly hindering the output of reflective light, thereby resulting in a decrease in reflectance and an increase in FWHM (21.1 nm). Despite these, both the TRMPC gel and solid exhibit brilliant colors and good monochromaticity. This characteristic is quite different from conventional PCM-PCs showing on-off colors under different temperatures[40–43]. The movement of the PCM in the TRMPC is supposed to be limited by the good affinity between PCM and MPC (Supplementary Fig. 12), leading to the slow and uniform crystallization of the PCM across the whole TRMPC and thus the bright color of the TRMPC solid. The thermal switching between the gel and solid states is highly reversible (Fig. 2e). It takes ≈1 s for TRMPC to finish the phase transition (Fig. 2f), suggesting the high thermoswitchable speed. The $\lambda$ and color of the TRMPC can be adjusted by altering the $\varphi_s$ of the MPC. As shown in Supplementary Fig. 13, the $\lambda$ of TRMPC decreases from 764 to 576 nm and the corresponding color changes from invisible to red and yellow when $\varphi_s$ increases from 15% to 40%.

The TRMPC exhibits significantly different mechanical properties in gel and solid states (Fig. 2g). The TRMPC gel is elastic and easily stretchable under low stress, showing a maximal stretching strain of 58% corresponding to a stress of 51.4 kPa. In dramatic contrast, during the stretching process, the TRMPC solid first experiences (1) reversibly elastic deformation (strain: 0–12%, maximal stress: 847.7 kPa), and (2) then irreversibly plastic deformations (strain: 12–118%, stress: 792–848 kPa). The Young's modulus of the TRMPC solid is 22.86 MPa, nearly 260 times higher than that of the TRMPC gel, firmly demonstrating that the deformability of the non-close-packing structure is greatly limited by the solid PCM at room temperature.

TRMPCs also can be fabricated by using long-chain-based PCMs. Here, other PCMs (Supplementary Fig. 14), including dodecanoic acid, myristic acid, palmitic acid, polyoxyethylene (23) lauryl ether (PLE), polyoxyethylene (20) cetyl ether (PCE), and polyoxyethylene (100) stearyl ether (PSE) were used to fabricate TRMPCs based on similar fabrication procedures. Compared to acids, TRMPCs were successfully fabricated when PLE, PCE, and PSE were used (Supplementary Fig. 15), which can be attributed to the different swelling behavior induced by their structure difference.

Like the typical PCM (named PNE), the PLE, PCE, and PSE also possess long chains and can be swelled into the MPC's polymer network with excellent swelling controllability. The swelling will be self-stopped after reaching the balance state, resulting in TRMPCs with the desired optical performance. Compared to PLE and PCE, the longer chain of PSE restricts the swelling, causing a smaller $\lambda$ and tuning range of wavelength ($\Delta\lambda = 112$ nm, Supplementary Fig. 16). Different from PLE, PCE, and PSE, the acid-based PCMs with short molecular lengths can be easily and rapidly swelled into the MPC's polymer network, leading to uncontrollable swelling and redshift of $\lambda$. Even worse, reflectance decreases quickly to ~0 during the swelling process, probably due to the break of long-range order by over-swelling. After cooling down, only white originating from the incoherent scattering of light by plenty of solid acids can be observed. These results suggest that PCMs with long-chain are suitable candidates for preparing TRMPCs and the responsive temperature can be tailored by simply using PCMs with different melting points.

## Thermoswitchable on-off mechanochromism

Compared to conventional MPCs, the TRMPC possesses a thermoswitchable on-off mechanochormism. When $T > T_m$, PCM melts, resulting in a TRMPC gel with highly sensitive mechanochromism (Fig. 3a). The pristine TRMPC gel displays a red color with a $\lambda$ located at

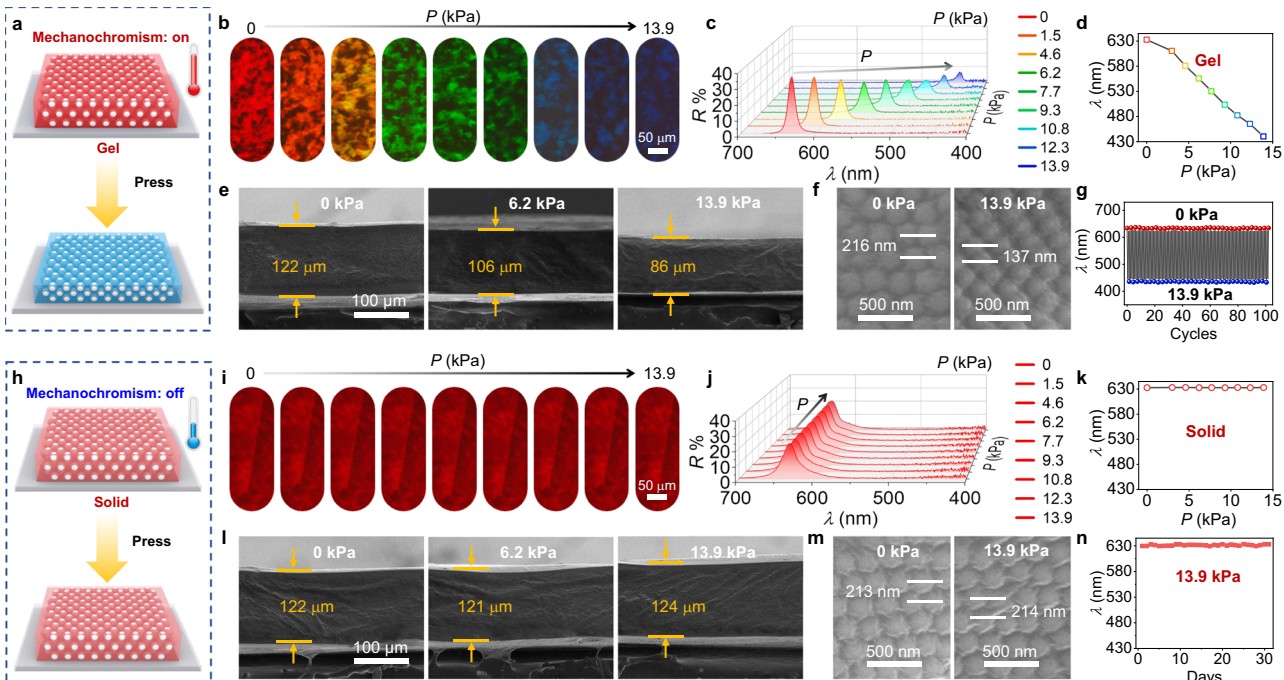

**Fig. 3 | Thermoswitchable on-off mechanochromism. a** Schematic illustration of the on-state mechanochromism of the TRMPC gel. **b** The microscope images and **c** reflection spectra of the TRMPC gel under different pressures. **d** The $\lambda$ of the TRMPC gel as a function of pressure. **e** The cross-sectional SEM images of TRMPC gel under the pressures of 0, 6.2, and 13.9 kPa, respectively. **f** The cross-sectional SEM images show the variation in lattice distance. **g** The $\lambda$ of the TRMPC gel as a function of alternate pressure and release. **h** Schematic illustration of the off-state mechanochromism of the TRMPC solid. **i** The microscope images and **j** reflection spectra of the TRMPC solid under different pressures. **k** The $\lambda$ of TRMPC solid as a function of pressure. **l** The cross-sectional SEM images of the TRMPC solid under the pressures of 0, 6.2, and 13.9 kPa, respectively. **m** The cross-sectional SEM images show the variation in lattice distance of the TRMPC solid. **n** The wavelength of the TRMPC solid as a function of time (under a constant pressure of 13.9 kPa).

632 nm. The structural color gradually changes from red to blue (Fig. 3b) and the corresponding $\lambda$ decreases from 632 to 440 nm (Fig. 3c, d) when the pressure increases from 0 to 13.9 kPa. The tuning range of the wavelength ($\Delta\lambda$) is 192 nm, similar to those of the best MPCs[44]. The sensitivity defined by $\Delta\lambda$/pressure (nm/kPa) is calculated to be 13.8 nm/kPa, higher than the most sensitive MPC[36,45,46]. The large $\Delta\lambda$ and high sensitivity can be attributed to increased $D_{id}$ caused by swelling, which leaves more room for deformation. The continuous decrease in $\lambda$ under pressure can be attributed to the gradual decrease in lattice distance (Fig. 3e). As presented in Fig. 3f, with a 13.9 kPa pressure loaded, the lattice distance decreases from 216 to 137 nm, which leads to the blueshift of $\lambda$ according to Bragg's law. The decrease in reflectance is due to the inevitable disturbance of the order degree by mechanical forces. After releasing pressure, the configuration and structural color of TRMPC gel return to the pristine state instantly. The switching of $\lambda$ under 100 cycles of alternate 0 and 13.9 kPa (Fig. 3g) demonstrates the excellent reversibility of the TRMPC gel. These results verify the on-state mechanochromism of the TRMPC gel.

Different from the gel, the TRMPC solid shows an off-state mechanochromism (Fig. 3h). When the $T < T_m$, PCM keeps a solid state, leading to poor deformability of the TRMPC solid. In the absence of pressure, the pristine TRMPC solid shows a red color (Fig. 3i), similar to that of the TRMPC gel. After being pressed from 0 to 13.9 kPa, both the structural color and $\lambda$ (Fig. 3j, k) of the TRMPC solid are barely changed. The solid PCM in the polymer network can efficiently resist external force, thereby preventing the deformation of the lattice distance (Fig. 3l, m) and thus the variation of color and $\lambda$. As presented in Fig. 3n, the TRMPC solid can retain its $\lambda$ over time, suggesting the off-state of mechanochromism. The adjustable and inactive colors respectively based on the on- and off-state mechanochromism of the TRMPC endow TRMPC with color recordability and erasability.

## Multicolor recordability and erasability

The color recordability was realized by pressing the TRMPC gel to obtain the desired color and subsequently color fixation by cooling down (Fig. 4a). At pristine state, the tree-like TRMPC gel (Fig. 4b) shows a brilliant red color and constant $\lambda$ of 632 nm in the absence of pressure. After loading pressure of 10.8 kPa at $T > T_m$, the color turns to blue, and the corresponding $\lambda$ changes to 482 nm due to the on-state mechanochromism. The blue color was then fixed by cooling down along with the gel-solid phase transition once $T < T_m$. After removing the pressure, interestingly, both the blue color and the corresponding $\lambda$ are retained over a long time (2 h) without returning to the pristine state. Recording of the changed color is attributed to the turning off of the TRMPC's mechanochromism at a deformed state. The recorded color can be erased by simply heating the deformed TRMPC solid to $T > T_m$. By heating, the blue TRMPC solid returns to the TRMPC gel displaying red color and $\lambda$, like the pristine one. It takes only 1.8 s to erase the recorded color (Fig. 4c), substantiating the fast speed of color erasing. Owing to the high $T_m$ (46 °C) of the PCM, the recorded color exhibits excellent stability under a wide range of temperatures (Fig. 4d).

The recorded color can be precisely adjusted by altering the pressure during the color generation process. As presented in Fig. 4e, f, desired $\lambda$ and colors ranging from red to yellow, green, and blue can be easily recorded by simply adjusting the pressure. The colors and corresponding $\lambda$ of these recorded colors at the solid state are similar to those at the gel state and can be retained for a long time (Supplementary Fig. 17), indicating the successful recording of diverse colors. Moreover, these recorded colors show constant hue and $\lambda$ under diverse pressure (Supplementary Fig. 18), further demonstrating that the TRMPC solid can resist external force by turning off the mechanochromic switch. These recorded colors can be easily erased by

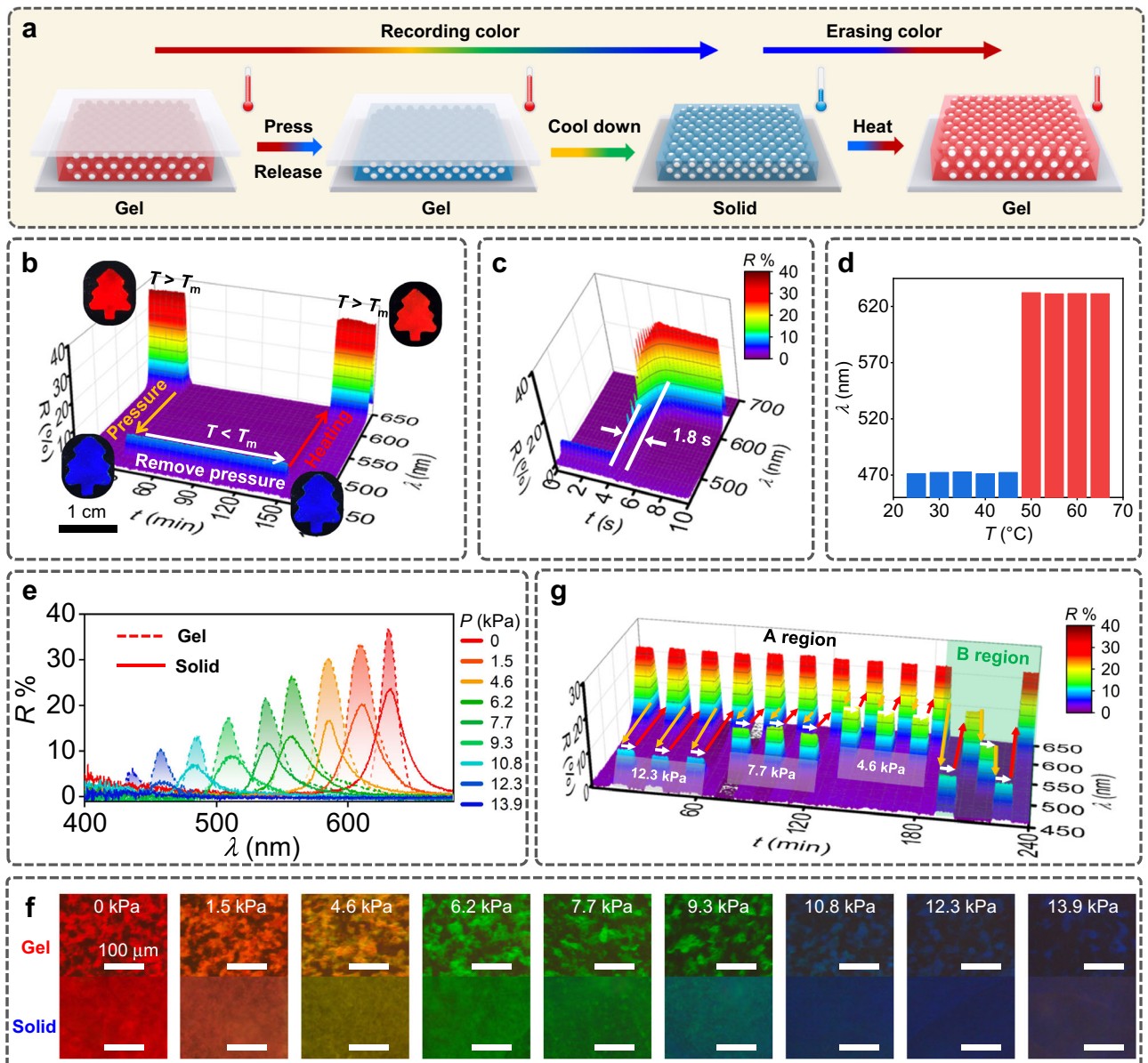

**Fig. 4 | Color recordability and erasability. a** Schematic illustration of the color recording and erasing of the TRMPC. **b** Recording a blue color on the red TRMPC gel by pressing (13.9 kPa) on and cooling down, and erasing the blue color by heating ($T > T_m$). **c** 3D reflection spectra show the variation of the reflection signal of the TRMPC during the heating process. **d** The $\lambda$ of the pressed TRMPC solid under different temperatures. **e** The reflection spectra and **f** the corresponding microscope images of the TRMPC gel and solid under different pressures. **g** The switch of reflective signals of the TRMPC by programmable color recordability and erasability.

heating the TRMPC solid to a $T > T_m$. In addition, these colors can be switched repeatedly or in a programmable way (Fig. 4g) by taking advantage of repeatable color recording and erasing. These results proved that TRMPCs possess repeatable multicolor recordability and erasability. The thermoswitchable on-off mechanochromism of the TRMPC is the key to these interesting characteristics; however, which is exceedingly difficult for conventional MPCs.

Further experimental results suggest that the percentage of the PCM plays an important role in color recordability. Here, the weight percentage of the PCM ($WP_{PCM}$) can be easily adjusted by altering the swelling time. As shown in Supplementary Fig. 19, $WP_{PCM}$ increases from 6.5% to 40.7% and the corresponding TRMPC's $\lambda$ increases from 509 to 630 nm can be easily obtained when adjusting the swelling time from 1 to 25 min, respectively. Then, these TRMPCs were pressed to a desired wavelength ($\lambda_1 = 450$ nm) at a $T > T_m$, followed by a cooling down process to record the blue color. Afterward, the pressure was removed and the corresponding wavelength ($\lambda_2$) was collected. In the

absence of PCM, the MPC returns to its pristine $\lambda$ instantly after releasing the pressure because of its elastic structure. When the $WP_{PCM}$ is low (6.5 and 12.0%, Supplementary Fig. 20a, b), the insufficient solid PCM cannot fully fix the deformed configuration of the TRMPC, leading to the partial recovery of $\lambda$ ($\lambda_1 < \lambda_2$) after releasing pressure. Thus, the targeted blue color cannot be recorded. When the $WP_{PCM}$ is high (24.5–40.7%, Supplementary Fig. 20c–f), the abundant solid PCM prevents the recovery of the compressed structure and thus retains the deformed configuration ($\lambda_1 = \lambda_2$). Thus, the targeted blue color was successfully recorded. These results indicate that a $WP_{PCM}$ higher than 24.5% is the key that endows TRMPC's color recordability. Compared to others, the TRMPC with the largest $\lambda$ of 630 nm and corresponding red color should be the optimized value due to its capability of recording more colors (Supplementary Fig. 21).

Multicolor recordability also can be realized by stretching the TRMPC at $T > T_m$, followed by cooling down till $T < T_m$. When the stretching strain increases from 0 to 51.2%, the structural color of the

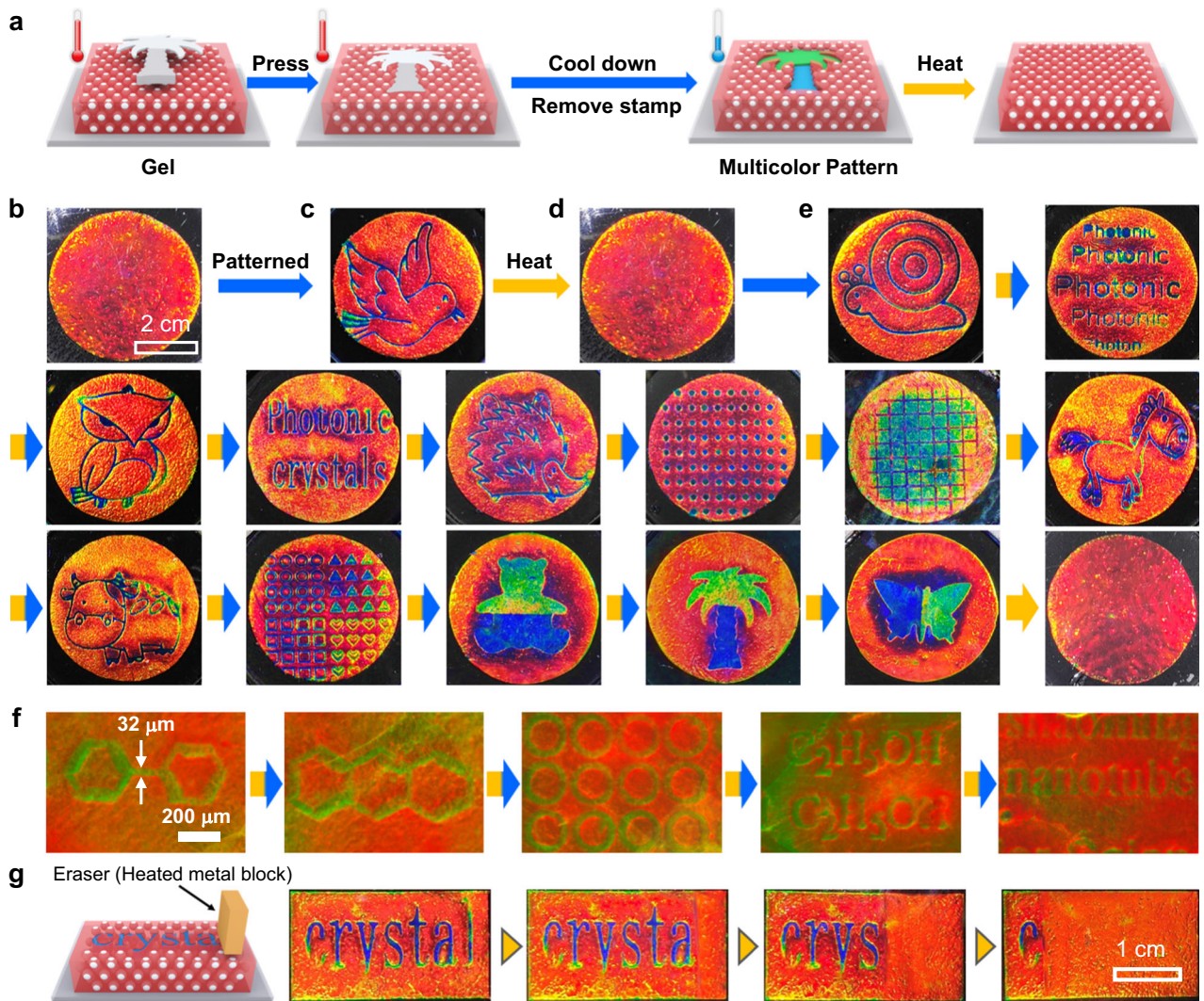

**Fig. 5 | Inkless rewritable papers. a** Schematic illustration of the processes of constructing and then reconfiguring patterns in an inkless way. **b–e** Digital photos of the **b** pristine, **c** patterned, **d** pattern-erased, and **e** pattern-reconfigured TRMPC papers. **f** The reconfiguration of high-resolution patterns. **g** Schematic illustration of erasing pattern selectively and the corresponding digital photos.

TRMPC gel turns from red to blue (Supplementary Fig. 22a), and the corresponding $\lambda$ blueshifts from 631 to 441 nm (Supplementary Fig. 22b, c). The tuning range of the wavelength ($\Delta\lambda$) by stretching is 190 nm, comparable to that (192 nm) by pressing. The sensitivity is 3.71 nm/%, much higher than those (1.11–3.16 nm/%) of highly sensitive MPCs. After cooling down, these colors can be recorded, as confirmed by their microscope images and $\lambda$. As expected, the recorded color can be erased by heating the TRMPC solid to $T > T_m$. These results proved that multicolor recordability can be realized by stretching during the color change process.

It should be noted that the multicolor recordability and erasability of the TRMPC are quite different from SMP-based structural color materials in the aspect of material fabrication, structure design, color-generating mechanism, optical properties, color recordability-erasability, and printing patterns (Supplementary Note 3 and Supplementary Table 1).

## Inkless rewritable papers

Conventional papers and prints have been widely used in our daily lives, especially in communication and information storage and spreading. Nevertheless, large-scale paper and ink consumption leads to significant environmental pollution, involving deforestation, toxic gases, water, and solids. Developing rewritable papers that can record information without using ink should be an ideal solution to address the above issues. In this work, the repeatable color recordability and erasability inspire us that the as-prepared TRMPC is indeed an excellent rewritable paper capable of constructing patterns in an inkless way.

In our design, the pattern was printed on a TRMPC by region-selective pressing the TRMPC gel using a stamp, followed by a cooling down process (Fig. 5a). At pristine, the TRMPC gel ($T > T_m$) shows a uniform red color (Fig. 5b) with a corresponding $\lambda$ located at 632 nm (Supplementary Fig. 23). Afterward, a stamp with a bird pattern was pressed on the TRMPC and a pressure of 12.3 kPa was applied to obtain a blue-colored pattern. The stamp was removed after cooling down ($T < T_m$), resulting in a blue bird on the red background (Fig. 5c) due to the on-off switch of mechanochromism. The $\lambda$ of the blue region is constant over half a year (Supplementary Fig. 24), demonstrating the good stability of the pattern. The pattern can be easily erased (Fig. 5d) by heating the patterned TRMPC to $T > T_m$ owing to the on-off switch of mechanochromism. The recovered TRMPC exhibits a similar red color and $\lambda$ to the pristine state, indicating a patterning-erasing cycle (Supplementary Movies 1–2) has a negligible effect on the optical properties of the TRMPC. The color recording and erasing can be

repeatedly realized more than 100 times (Supplementary Fig. 25). Using similar procedures, a variety of multicolor (Fig. 5e) and high-resolution (Fig. 5f) patterns can be repeatedly reconfigured on the same TRMPC. Supplementary Fig. 26 suggests that the minimum pressure for generating an effective pattern should be around 2.2 kPa. The line and point resolutions that we can obtain are measured to be 7.2 and 5.4 μm (Supplementary Fig. 27), respectively. We believe that better resolution can be realized with more elaborate stamps. In addition, the pattern can be selectively erased through heating the desired region. As presented in Fig. 5g, a crystal pattern was firstly printed on the TRMPC. The l of the crystal was selectively erased by simply covering a preheated copper metal ($T > T_m$) on it. Similarly, each letter can be removed in a programmable way.

Compared to conventional inkless rewritable papers (CIRPs) based on the oxide and redox of organic dyes[47–50], TRMPC paper possesses various advantages: (1) multicolor patterns can be easily achieved using TRMPC paper, while only mono-color patterns can be obtained based on CIRPs; (2) TRMPC paper-based pattern exhibits anti-photobleaching and stable colors, in dramatic contrast to the CIRP-based pattern suffering from photobleaching; and (3) the printing of pattern is considerably efficient and convenient without requiring complex instrument. The utility of TRMPC paper instead of conventional papers will facilitate not only the reduction of environmental pollution but also the applications in green printing, sensors, optical devices, and so forth.

In summary, TRMPCs capable of recording and erasing multicolor were fabricated by (1) self-assembling silica particles into acrylates, followed by a photopolymerization process to achieve MPC with non-close-packing structures and subsequently (2) swelling the MPC into the liquid PCM with a high $T_m$. The loading of PCM into the polymer network of MPC endows TRMPC $T$-dependent on-off mechanochromism along with the solid-gel phase changing, contributing to the color recordability and erasability, respectively. The color recording was realized by two processes: color generation by pressing or stretching the TRMPC with on and highly sensitive mechanochromism at $T > T_m$ and color fixation by cooling down the pressed TRMPC to turn off the mechanochromism till $T < T_m$. The generated color can be precisely adjusted from red to blue by increasing the pressure (0–13.9 kPa). The external force-induced compression of the lattice distance of the non-close-packing structure is the key to color generation. The fixation of the compressed configuration of TRMPC by the solid PCM is crucial to the color fixation. In contrast, the recorded color can be easily erased by heating the deformed TRMPC to $T > T_m$ that allows the recovery of the TRMPC to the pristine (relax) state. The color recordability and erasability make TRMPC an ideal candidate as an inkless rewritable paper, on which multicolor color and high-resolution patterns have been repeatedly printed. Compared to conventional papers, the usage of TRMPC paper will significantly reduce deforestation and environmental pollution thanks to no use of inks and disposable papers. The on-off thermoswitchable mechanochromism will facilitate the applications of TRMPC in green printing, low-power display, and next-generation sensors.

## Methods
### Materials
Di(ethylene glycol) ethyl ether acrylate (DEGEEA), polyethylene glycol mono-phenyl ester acrylate (PEGPEA), poly(propylene glycol) acrylate (PPGA), poly(ethylene glycol) diacrylate (PEGDA, $M_n$: 250), pentaerythritol tetraacrylate (PETA), polyoxyethylene (40) nonylphenyl ether (PNE, the typical PCM, $C_{15}H_{24}O(C_2H_4O)_n$, $M_n$: 1982), polyoxyethylene (23) lauryl ether (PLE, $(C_2H_4O)_nC_{12}H_{26}O$, $M_n$: 1198), polyoxyethylene (20) cetyl ether (PCE, $HO(CH_2CH_2O)_{20}C_{16}H_{33}$, $M_n$: 1124), polyoxyethylene (100) stearyl ether (PSE, $C_{18}H_{37}(OCH_2CH_2)_nOH$, $M_n$: 4670), dodecanoic acid ($CH_3(CH_2)_{10}COOH$), myristic acid ($CH_3(CH_2)_{12}COOH$), palmitic acid ($CH_3(CH_2)_{14}COOH$), and 2-hydroxy-2-methylpropiophenone (photo-

initiator, 96%) were obtained from Sigma-Aldrich. 1-Ethyl-3-methylimidazolium bis(trifluoromethylsulfonyl)imide (EMBTF, 97%) was purchased from Aladdin. All the chemicals were used as received without further purification.

### Preparation of MPC
Briefly, silica particle powders (167 nm, 0.03 cm³) were dispersed in a mixture of ethanol (0.5 mL), PEGPEA (0.021 mL), and DEGEEA (0.049 mL) containing 5% photo-initiator by sonication. The mixed solution was heated in an oven at 100 °C for 1 h and approximately 0.1 mL precursor solution showing iridescent color was obtained. This precursor solution (0.03 mL) was then sandwiched between two substrates separated by 0.09 mm and converted into a green MPC after irradiating by UV light (365 nm, 4.8 mW cm⁻²) for 3 min. The distance between the UV light source and precursor solution is about 15 cm.

### Fabrication of TRMPC
Typically, the red TRMPC was achieved by swelling the green MPC with excessive PCM (5 mL) at 80 °C for 1 h to reach the swelling balance state and wiping off the redundant PCM. The TRMPC is an organic gel at $T > T_m$ but solid at $T < T_m$. The reflection wavelength of the TRMPC can be controlled by adjusting the swelling times. TRMPCs with reflection peak positions located at 513, 541, 569, 601, and 630 nm respectively can be obtained when the swelling time is adjusted to 2, 6, 15, 20, and 25 min, respectively.

### Printing and then reconfigure PC patterns
Firstly, the red TRMPC gel was heated on the hot plate (80 °C). Secondly, the stamp with a preset pattern was pressed on the red TRMPC gel to generate patterns which were then fixed by cooling down to room temperature and removing the stamp. The color of the patterned region can be adjusted by pressure. Typically, red, yellow, green, and blue colors can be achieved when 0, 4.6, 7.7, and 12.3 kPa pressure are used, respectively. The pattern can be erased by heating the TRMPC on the hot plate (80 °C) for 1 min to release the stress.

### Characterizations
The structures of TRMPCs were investigated using Hitachi SEMSU8010. The elastic properties of TRMPCs were measured using an electronic universal testing machine (Hegewald & Peschke, Inspekt Table Blue 5kN). The accuracy of the strain measurement is 0.1%. For most cases, the reflection spectra of TRMPCs are obtained by fixing a collection angle to 0°. The reflectance spectra were measured using a NOVA spectrometer (Hamamatsu, S7031). Angle-resolved spectra are collected by changing reflection angles from 0 to 60° using the angle-resolved spectrum system (R1, Ideaoptics, China) equipped with a highly sensitive spectrometer (NOVA, Ideaoptics, China). The optical microscope images and microscopic reflectance spectra were obtained on an Olympus BXFM reflection-type microscope operated in the darkfield mode.

## Data availability
The data that support the findings of this study are available within the paper and Supplementary Information. Additional relevant data are available from the corresponding author on request.

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

## Acknowledgements

This work was financially supported by the National Natural Science Foundation of China (52302001, 51920105004, 12075154, 21673160, and 52371196), the Natural Science Foundation of Guangdong Province (2023A1515010063) and Startup Foundation of Shaoxing University (13011001002/261).

## Author contributions

Y. Hu performed the studies and conducted the experiments. C. Z. Qi and D. K. Ma contributed to the data analysis. D. P. Yang and S. M. Huang supervised this work. All authors discussed the results and contributed to the writing and revision of the manuscript.

## Competing interests

The authors declare no competing interests.
