## [Peer Review File · Nature Communications]

Multicolor recordable and erasable photonic crystals based on on-off thermoswitchable mechanochromism toward inkless rewritable paperREVIEWER COMMENTS

Reviewer #1 (Remarks to the Author):

This article proposes a novel strategy and a new material. The new type of thermal-responsive MPC (TRMPC) with a unique multicolor recordability and erasability was fabricated by combining non-closely packed MPCs and phase-change materials (PCM). However, many problems remain in the article and require major revisions. If the following problems can be perfectly solved, I recommend that the article be published in Nature Communications.

1. Δn is defined in this article as the difference between the refractive index of SiO₂ and the refractive index of the mixed polymer (DEGEEA+PEGPEA). However, there are conflicting statements regarding the effect of Δn on reflectivity. For example, “low Δn ” appears in line 76, 135, and 168, resulting in high reflectivity, however, “large Δn ” appears in line 197 and 149, resulting in high reflectivity. The author should explain this contradictory statement.
2. When “refractive index contrast” is mentioned for the first time in line 76, its definition is not directly specified, which may cause readers to mistakenly think that it is the contrast between MPC and PCM or the contrast between DEGEEA and PEGPEA. The author should clarify this issue.
3. The highlight of this article is the use of the solid-gel phase transition characteristic of PCM. Are there other materials with similar property? It is mentioned in line 210 that the introduction of PCM will lead to a slight increase in Δn . Will this have the negative effects of enhanced incoherent scattering and the break of long-range order mentioned earlier? The reason for the increase in Δn should be provided.
4. Only Figure 2 shows the bonding configuration of PCM. The author should give specific chemical component of PCM.
5. TRMPC transforms from solid to gel when $T > T_m$, and the color changes (such as red to blue) when pressure is applied. After cooling to $T < T_m$, the blue color remains unchanged when the pressure is removed (color recording), and finally heated to $T > T_m$, returning to red. Therefore, I propose to add schematic arrows for “Cool down” and “Color recordability” (pointing from right to left) between the two blue models in Figure 1.
6. More experiments on “off” mechanochromism should be added. After cooling and solidification to form a blue solid, pressure can be applied to it, further proving that the on-off switch can resist external force as color recording, and this effect can also be illustrated in Figure 1.
7. The author mentioned that both the order degree and Δn affect reflectivity. However, in the two paragraphs from line 180 to 197, the author used changes in reflectivity to illustrate the impact of Δn .

on the order degree and f_p on Δn respectively. The effect of $\Delta n(f_p)$ on reflectivity should be discussed without the effect of $f_p(\Delta n)$.

8. The optimization conditions have been given in the first two paragraphs of Section 3.1, is $\phi_d:\phi_p=7:3$, $\Delta n=0.025$. However, it is not until the two paragraphs from line 180 to 197 that the effect of $\phi_d:\phi_p$ ($f_d:f_p$) and Δn on reflectivity begins to be analyzed. This order of discussion does not seem to be logical.

9. Equations (1)-(4) are all mentioned in the text, but only (1) is marked. The marks of Figure 3 are incorrectly positioned in the text. For example, Figure 3(m, n) should correspond to line 253 “preventing the deformation of the lattice distance” and Figure 3(l) should correspond to line 255 “retain its λ under much higher pressure”, etc. Figure 1 is not marked in the text. The author should carefully correct other errors like this.

Reviewer #2 (Remarks to the Author):

This manuscript reports on multicolor recordable and erasable photonic crystals that are integrated with phase-change materials (PCM). These materials enable the regulation of the structural properties through reversible transitions between solid and gel states, driven by temperature changes. The mechanochromic photonic crystals (MPCs), consist of non-closed packed silica colloid arrays embedded within a PEGPEA and DEGEEA matrix. The thermal-responsive MPCs (TRMPCs) are crafted by immersing the MPCs in PCM, inducing swelling at temperatures above the PCM's T_m . This process allows for multicolor recordability through mechanical pressure applied in the gel state of TRMPCs, with subsequent color fixation achieved by cooling the TRMPCs to a solid state below T_m . The deformed colors revert to their initial state upon reheating the TRMPCs above T_m . This method demonstrates the capability to record colors with exceptional stability and reconfigure them within a short period simply by applying heat. The incorporation of PCM into elastic photonic materials is interesting and the results are well organized. However, similar working principle of recordable and erasable photonic crystals has been demonstrated with shape memory polymer and the advancement or novelty of this work are not sufficient to guarantee the publication in Nature communications. Hope the following comments help the authors improve their work.

1. The incorporation of photonic crystals with phase-change materials in this research, enabling permanent and erasable multicolor recording, is notable. However, it utilizes a methodology similar to that of shape memory polymer-based research (Adv. Optical Mater. 2021, 9, 210073), which involves applying pressure above the glass transition temperature (T_g), then cooling to fix the color recording, and reheating above T_g to recover the original state. It would be beneficial to further articulate how the approach of this study distinctively diverges from earlier research.

2. To efficiently achieve multicolor recording in a single operation, managing strain instead of

applying pressure is considered to be more effective. Accordingly, it would be greatly appreciated if a strain-stress curve could be provided, illustrating how strain varies in response to stress. In this regard, providing a strain-stress curve that shows the change in strain according to stress would be very useful.

3. For the practical application of this inkless rewritable paper, resolution is an important factor. It is imperative to ascertain the minimal pressure necessary for effective patterning, as well as to delineate the achievable resolution of these patterns. This understanding is crucial in evaluating the paper's practical utility and functional performance.

Reviewer #3 (Remarks to the Author):

I co-reviewed this manuscript with one of the reviewers who provided the listed reports.

Response to reviewers

Reviewers 1 and 3 have co-reviewed this work: *This article proposes a novel strategy and a new material. The new type of thermal-responsive MPC (TRMPC) with a unique multicolor recordability and erasability was fabricated by combining non-closely packed MPCs and phase-change materials (PCM). However, many problems remain in the article and require major revisions. If the following problems can be perfectly solved, I recommend that the article be published in Nature Communications.*

Author reply: Thank you very much for your high praise and positive summary of this work. We are very grateful for your insightful comments and valuable suggestions, which can greatly help us to improve the work. Following your comments, we have carefully revised the manuscript accordingly (please see the highlighted part in red in the revised manuscript) and provided concrete responses point-by-point as follows.

(1) Δn is defined in this article as the difference between the refractive index of SiO_2 and the refractive index of the mixed polymer (DEGEEA+PEGPEA). However, there are conflicting statements regarding the effect of Δn on reflectivity. For example, “low Δn ” appears in line 76, 135, and 168, resulting in high reflectivity, however, “large Δn ” appears in line 197 and 149, resulting in high reflectivity. The author should explain this contradictory statement.

Author reply: Thank you very much for your insightful comment. For the MPC, the scattering and reflectance originating from incoherent and coherent scattering of light, respectively, are proportional to Δn . When Δn is large, both scattering and reflectance are intense. Decreasing Δn will improve the structural color purity by suppressing the incoherent scattering of light, despite the slight decrease in reflectance. However, when Δn is close to 0, the incoherent light scattering is extremely weak while reflectance is still highly sensitive to Δn . The decrease in Δn will lead to a dramatic decrease in reflectance and color saturation, which is not favorable in practical applications. In this regard, the silica/PEGPEA/DEGEEA MPC should possess an appropriately small Δn that enables high reflectance, weak scattering, and thus good color purity. The refractive index (n) of the silica particles (n_s), PEGPEA (n_p), and DEGEEA (n_d) are 1.460, 1.520, and 1.470, respectively. PEGPEA with a high n is used to increase Δn , which can be calculated by equations 1-2, where ϕ_p and ϕ_d are the volume fractions of PEGPEA and DEGEEA, respectively. With f_p increases from

0 to the optimized 0.3, Δn increases from 0.010 to 0.025, leading to an increase in reflectance from 19% to 47%. Despite the increase in Δn , the optimized $\Delta n = 0.025$ is still small enough to efficiently suppress the incoherent light scattering. Thus, the optimized MPC possesses both high reflectance, brilliant color, and negligible scattering.

The “Large Δn ” in lines 149 and 197 means that the Δn of the optimized MPC ($f_p = 0.3$) is larger than other MPCs with different f_p , which causes misunderstanding and confusion. The “Large Δn ” has been replaced by “appropriate Δn ” in the revised manuscript.

$$\Delta n = n_{\text{silica}} - n_{\text{surroundings}} \quad (1)$$

$$n_{\text{surroundings}} = (n_p \phi_p + n_d \phi_d) / (\phi_p + \phi_d) \quad (2)$$

The related revision can be found in lines 100 and 129 in the revised manuscript.

(2) When “refractive index contrast” is mentioned for the first time in line 76, its definition is not directly specified, which may cause readers to mistakenly think that it is the contrast between MPC and PCM or the contrast between DEGEEA and PEGPEA. The author should clarify this issue.

Author reply: I am sorry for the mistake. Refractive index contrast (Δn , calculated by eq 1) is the index difference between silica particles and surroundings, which has been corrected in the revised manuscript.

$$\Delta n = n_{\text{silica}} - n_{\text{surroundings}} \quad (1)$$

The related revision is in lines 79-80 in the revised manuscript.

(3) The highlight of this article is the use of the solid-gel phase transition characteristic of PCM. Are there other materials with similar properties? It is mentioned in line 210 that the introduction of PCM will lead to a slight increase in Δn . Will this have the negative effects of enhanced incoherent scattering and the break of long-range order mentioned earlier? The reason for the increase in Δn should be provided.

Author reply: Thank you very much for your kind comment. TRMPCs also can be fabricated by using long-chain-based PCMs. Except for the typical PCM, other PCMs (Fig. R1), including

dodecanoic acid, myristic acid, palmitic acid, polyoxyethylene (23) lauryl ether (PLE), polyoxyethylene (20) cetyl ether (PCE), and polyoxyethylene (100) stearyl ether (PSE) were used to fabricate TRMPCs based on similar fabrication procedures. Compared to acids, TRMPCs were successfully fabricated when PLE, PCE, and PSE were used (Fig. R2), which can be attributed to the different swelling behavior induced by their structure difference.

Like the typical PCM (named PNE), the PLE, PCE, and PSE also possess long chains and can be swelled into the MPC's polymer network with excellent swelling controllability. The swelling will be self-stopped after reaching the balance state, resulting in TRMPCs with desired optical performance. Compared to PLE and PCE, the longer chain of PSE restricts the swelling, causing a smaller λ and tuning range of wavelength ($\Delta\lambda = 112$ nm, Fig. R3). Different from PLE, PCE, and PSE, the acid-based PCMs with short molecular lengths can be easily and rapidly swelled into the MPC's polymer network, leading to uncontrollable swelling and redshift of λ . Even worse, reflectance decreases quickly to ~ 0 during the swelling process, probably due to the break of long-range order by over-swelling. After cooling down, only white originating from the incoherent scattering of light by plenty of solid acids can be observed. These results suggest that PCMs with long-chain are suitable candidates for preparing TRMPCs and the responsive temperature can be tailored by simply using PCMs with different melting points.

Fig. R1 Chemical structure of different PCMs. The chemical structure of **a** dodecanoic acid ($\text{CH}_3(\text{CH}_2)_{10}\text{COOH}$), **b** myristic acid ($\text{CH}_3(\text{CH}_2)_{12}\text{COOH}$), **c** palmitic acid ($\text{CH}_3(\text{CH}_2)_{14}\text{COOH}$), **d** polyoxyethylene (23) lauryl ether (PLE, $(\text{C}_2\text{H}_4\text{O})_n\text{C}_{12}\text{H}_{26}\text{O}$), **e** polyoxyethylene (20) cetyl ether (PCE, $\text{HO}(\text{CH}_2\text{CH}_2\text{O})_{20}\text{C}_{16}\text{H}_{33}$), and **f** polyoxyethylene (100) stearyl ether (PSE, $\text{C}_{18}\text{H}_{37}(\text{OCH}_2\text{CH}_2)_n\text{OH}$), respectively.

Fig. R2 TRMPCs with diverse PCMs. The reflection spectra and corresponding digital photos of TRMPCs with different PCMs, **a** dodecanoic acid, **b** myristic acid, and **c** palmitic acid, **d** PLE, **e** PCE, and **f** PSE.

Fig. R3 Mechanochromic properties of the TRMPC gel prepared by PSE under different pressures. a The reflection spectra and **b** the corresponding microscope images of the TRMPC gel under different pressures.

The related revision can be found in lines 202-220 in the revised manuscript.

The Δn of the TRMPC was calculated by equations 1 and 2. The refractive index (n) of the silica particles (n_s), PEGPEA (n_p), DEGEEA (n_d), and PCM (n_{PCM}) are 1.460, 1.520, 1.470, and 1.505, respectively. For TRMPC, the volume fractions of silica particles (ϕ_s), PEGPEA (ϕ_p), DEGEEA (ϕ_d), and PCM (ϕ_{PCM}) are 13.2%, 9.3%, 21.6%, and 55.9%, respectively. The Δn of TRMPC is calculated to be 0.038 according to equations 1-2. In comparison, the ϕ_s , ϕ_p , and ϕ_d of the MPC are 30%, 21%, and 49%, respectively, resulting in a $\Delta n = 0.025$ calculated by equations 1 and 3. Therefore, the introduction of the PCM with high n into the MPC leads to a slight increase in Δn .

$$\Delta n = n_{\text{silica}} - n_{\text{surroundings}} \quad (1)$$

$$\text{For TRMPC, } n_{\text{surroundings}} = (n_p\phi_p + n_d\phi_d + n_{PCM}\phi_{PCM}) / (\phi_p + \phi_d + \phi_{PCM}) \quad (2)$$

$$\text{For MPC, } n_{\text{surroundings}} = (n_p\phi_p + n_d\phi_d) / (\phi_p + \phi_d) \quad (3)$$

The related revision can be found in Supplementary Note 1 in the revised supporting information manuscript.

Despite the slight increase in Δn , the TRMPC exhibits no obvious enhancement in incoherent light scattering compared to the MPC (Fig. R4). This can be explained by that the TRMPC's Δn (0.038) is still small, which can efficiently suppress the incoherent scattering of light. Fig. R5 shows that MPC and TRMPC show highly ordered structures without significant differences in

order degree, demonstrating that the order degree is well retained with negligible influence by the PCM.

Fig. R4 Comparison of the incoherent scattering of light between the MPC, TRMPC gel, and TRMPC solid. a-c The scattering spectra of the MPC, TRMPC gel, and TRMPC solid. The incident angle is fixed to 0° and the detection angle varies from 10° to 60° with an interval of 5° .

Fig. R5 Comparison of the order degree between the MPC and TRMPC. a-b SEM images of MPC and TRMPC, respectively.

The related revision can be found in lines 175-180 in the revised manuscript.

(4) Only Figure 2 shows the bonding configuration of PCM. The author should give specific chemical component of PCM.

Author reply: We are grateful for your kind suggestion. In this work, polyoxyethylene (40) nonylphenyl ether (PNE, $C_{15}H_{24}O(C_2H_4O)_n$, M_n : 1982) is used as the PCM.

The related revision can be found in lines 78 and 367 (experimental section) in the revised manuscript.

(5) TRMPC transforms from solid to gel when $T > T_m$, and the color changes (such as red to blue) when pressure is applied. After cooling to $T < T_m$, the blue color remains unchanged when the pressure is removed (color recording), and finally heated to $T > T_m$, returning to red. Therefore, I propose to add schematic arrows for “Cool down” and “Color recordability” (pointing from right to left) between the two blue models in Figure 1.

Author reply: Thank you very much for your valuable suggestion. “Cool down” and “Color recording” have been added into the Fig. 1 (red rectangle) as follows.

Fig. 1 Schematic illustration of the working mechanism of the TRMPC. TRMPC transforms from solid to gel when $T > T_m$ and its color changes (such as red to blue) when pressure is applied. After cooling to $T < T_m$, the blue color remains unchanged when the pressure is removed (color recordability), and finally returns to red after being heated to $T > T_m$.

The related revision can be found in lines 533-537 in the revised manuscript.

(6) More experiments on “off” mechanochromism should be added. After cooling and solidification to form a blue solid, pressure can be applied to it, further proving that the on-off switch can resist external force as color recording, and this effect can also be illustrated in Figure 1.

Author reply: Thanks a lot for your kind suggestion. As shown in Fig. R6, TRMPC solids with different λ (440-610 nm) and colors (from blue to red) have been fabricated by applying different

pressures during the color change processes. Then, external pressure was applied to these TRMPC solids and the corresponding λ and microscope images were collected. As expected, the color and λ of these TRMPC solids remain constant under different pressures, further verifying that TRMPCs can resist external forces by turning off the mechanochromic switch.

Fig. R6 The “off” mechanochromism of the TRMPC solid under different pressures. **a** The microscope images of TRMPCs under different pressures. **b** The λ of TRMPCs solid as a function of pressure.

The related revision can be found in lines 267-269 in the revised manuscript.

The effect that the blue TRMPC solid can resist external force has been illustrated in Fig. 1 (green rectangle).

Fig. 1 Schematic illustration of the working mechanism of the TRMPC. TRMPC transforms from solid to gel when $T > T_m$ and its color changes (such as red to blue) when pressure is applied. After cooling to $T < T_m$, the blue color remains unchanged when the pressure is removed (color recordability), and finally returns to red after being heated to $T > T_m$.

The related revision can be found in lines 533-537 in the revised manuscript.

(7) The author mentioned that both the order degree and Δn affect reflectivity. However, in the two paragraphs from lines 180 to 197, the author used changes in reflectivity to illustrate the impact of f_p on the order degree and f_p on Δn respectively. The effect of $\Delta n(f_p)$ on reflectivity should be discussed without the effect of $f_p(\Delta n)$.

Author reply: Thanks for very much your insightful comment. We first investigate the effect of Δn on reflectivity without changing f_p . The ϕ_s , ϕ_p , and ϕ_d of the MPC are set as 30%, 21%, and 49%, respectively, with $f_p = 0.3$ and $f_d = 0.7$. The refractive index (n) of the silica particles (n_s), PEGPEA (n_p), and DEGEEA (n_d) are 1.460, 1.520, and 1.470, respectively. The Δn of the MPC can be calculated by equations 1-2. Then, DEGEEA was replaced by other acrylates, including PPGA ($n = 1.456$, $\Delta n = 0.015$), DEGEEA ($n = 1.471$, $\Delta n = 0.026$), PEGDA ($n = 1.506$, $\Delta n = 0.050$), and PETA ($n = 1.527$, $\Delta n = 0.065$) to prepare MPCs with the same f_p (0.3) but different Δn . The SEM images of as-fabricated MPCs with diverse Δn show similar highly ordered structures (Fig. R7a), indicating the change in Δn has a negligible effect on the structural order. As shown in Fig. R7b-c, the reflectance of MPCs increases from 15% to 68% when Δn increases from 0.015 to 0.065, demonstrating that the reflectance is proportional to the Δn . Besides, the λ also increases slightly as Δn increases, which can be attributed to the slight increase in the effective n of MPCs.

$$\Delta n = n_{\text{silica}} - n_{\text{surroundings}} \quad (1)$$

$$\text{For MPC, } n_{\text{surroundings}} = (n_p \phi_p + n_d \phi_d) / (\phi_p + \phi_d) = n_p f_p + n_d f_d \quad (2)$$

Fig. R7 The effect of Δn on the MPC's reflectance. **a** SEM images of MPCs with different Δn . **b** Reflection spectra of MPCs with different Δn . **c** The reflectance and λ of MPCs as a function of Δn .

We then investigate the effect of f_p on the reflectance without altering Δn . For the silica/PEGPEA/DEGEEA MPC, it is impossible to change the f_p only without influencing its Δn according to equations 1 and 3. In principle, we can fabricate the silica/PEGPEA MPC, and thus investigate the influence of f_p without changing Δn , because the silica particles are surrounded by only PEGPEA ($\varphi_p = f_p$). For the silica/PEGPEA MPC, its Δn is always a constant value (0.06) regardless of φ_p . As shown in Fig. R8, when $\varphi_p = 0.78$ and 0.80, the corresponding MPCs exhibit amorphous structure, leading to negligible reflectance and structural color. The amorphous structures originate from the weak electrostatic repulsion between silica particles caused by the large interparticle distance. When φ_p gradually decreases to 0.72, the interparticle distance decreases, leading to the increase in electrostatic repulsion, order degree, and hence reflectance. As $\varphi_p = 0.70$, the reflectance is similar to that of $\varphi_p = 0.72$ owing to their similar order degree. Therefore, the reflectance of the silica/PEGPEA MPC is also proportional to the φ_p -dependent order degree. The decrease in φ_p will cause the increase in interparticle distance and thus the decrease in λ according to Bragg's law.

The above results demonstrated that reflectance is positively dependent on Δn and φ_p -based order degree. A large Δn and high-order degree will result in a high reflectance.

Fig. R8 Effect of φ_p on the reflectance of MPCs. a-f SEM images, digital photos, and reflection spectra of MPCs with different φ_p . The diameter of all the samples is 1 μm .

Due to the limitation in the number of words, these results and discussion have been removed to the supporting information. The related revision can be found in lines 111-114 in the revised manuscript and Supplementary Note 2 in the supporting information.

(8) The optimization conditions have been given in the first two paragraphs of Section 3.1, is $\varphi_d:\varphi_p=7:3$, $\Delta n = 0.025$. However, it is not until the two paragraphs from lines 180 to 197 that the effect of $\varphi_d:\varphi_p$ ($f_d:f_p$) and Δn on reflectivity begin to be analyzed. This order of discussion does not seem to be logical.

Author reply: Thank you for this valuable feedback. The two paragraphs from lines 180 to 197 were moved to the 2-3 paragraph in the “Results” part to make the discussion more logical.

The related revision can be found in lines 115-1129 in the revised manuscript.

(9) Equations (1)-(4) are all mentioned in the text, but only (1) is marked. The marks of Figure 3 are incorrectly positioned in the text. For example, Figure 3(m, n) should correspond to line 253 “preventing the deformation of the lattice distance” and Figure 3(l) should correspond to line 255 “retain its λ under much higher pressure”, etc. Figure 1 is not marked in the text. The author should carefully correct other errors like this.

Author reply: I am very sorry for these mistakes. The “eq” was used to represent the “equation” in our original manuscript, and we have corrected these mistakes in our revised manuscript.

Equations 1, 2-3, and 4 can be found in lines 79-80, 125, and 143 in the revised manuscript.

The mismatch between Figure 3 (m-l) and the main text is due to the mislabels in Fig. 3 (old version, blue circles), which has been corrected in the revised manuscript (blue cycles). Figure 1 was added in the third paragraph of the introduction in the revised manuscript.

Fig. 3 Old version with mislabeled l-n.

Fig. 3 Thermo-switchable on-off mechanochromism. **h** Schematic illustration of the “off” mechanochromism of the TRMPC solid; **i** The microscope images and **j** reflection spectra of the TRMPC solid under different pressures; **k** The λ of TRMPC solid as a function of pressure; **l** The cross-sectional SEM images of the TRMPC solid under the pressures of 0, 6.2, and 13.9 kPa, respectively; **m** The cross-sectional SEM images show the variation in lattice distance of the TRMPC solid; **n** The wavelength of the TRMPC solid as a function of time (under a constant pressure of 13.9 kPa).

The revised Fig. 3 can be found in lines 550-562 in the revised manuscript.

Reviewer #2: *This manuscript reports on multicolor recordable and erasable photonic crystals that are integrated with phase-change materials (PCM). These materials enable the regulation of the structural properties through reversible transitions between solid and gel states, driven by temperature changes. The mechanochromic photonic crystals (MPCs), consist of non-closed packed silica colloid arrays embedded within a PEGPEA and DEGEEA matrix. The thermal-responsive MPCs (TRMPCs) are crafted by immersing the MPCs in PCM, inducing swelling at temperatures above the PCM's T_m . This process allows for multicolor recordability through mechanical pressure applied in the gel state of TRMPCs, with subsequent color fixation achieved by cooling the TRMPCs to a solid state below T_m . The deformed colors revert to their initial state upon reheating the TRMPCs above T_m . This method demonstrates the capability to record colors with exceptional stability and reconfigure them within a short period simply by applying heat. The incorporation of PCM into elastic photonic materials is interesting and the results are well organized. However, similar working principle of recordable and erasable photonic crystals has been demonstrated with shape memory polymer and the advancement or novelty of this work are not sufficient to guarantee*

the publication in Nature Communications. Hope the following comments help the authors improve their work.

Author reply: Thank you very much for your positive comments on this work. We are very grateful for your insightful and valuable suggestions, which provide us with many valuable directions to revise the manuscript. Following your comments, we have carefully revised the manuscript accordingly (please see the highlighted part in red in the revised manuscript) and provided concrete responses point-by-point as follows.

(1) The incorporation of photonic crystals with phase-change materials in this research, enabling permanent and erasable multicolor recording, is notable. However, it utilizes a methodology similar to that of shape memory polymer-based research (Adv. Optical Mater. 2021, 9, 210073), which involves applying pressure above the glass transition temperature (T_g), then cooling to fix the color recording, and reheating above T_g to recover the original state. It would be beneficial to further articulate how the approach of this study distinctively diverges from earlier research.

Author reply: We greatly appreciate the reviewer's comments and fully understand why the concern was raised. In the initial submission, we very regret that the novel principle of this work is not fully demonstrated. In fact, our work is quite different from reference 1 (Adv. Optical Mater. 2021, 9, 2100739) in the aspect of materials, structures, optical properties, and working mechanisms of color recordability and erasability, which is listed in Table R1.

Table R1. Comparisons of materials, structures, and properties between reference 1 and our work.

	Ref 1 (SMRSCF)	Our work (TRMPC)
Materials	Epoxy resin/PU/PA	SiO ₂ /PEGPEA/DEGEEA/PCM
Structure	Closely arranged micro-dome monolayer	Non-close-packing ordered structures
Color-generating mechanism	The combined effect of total internal reflection (TIR) and interference	Light diffraction by ordered structures
Structural color	Retroreflection	Reflection
Color purity	Poor	Good

	Colorless recording-erasing based on shape memory effect	Multicolor recording-erasing based on thermoswitchable on-off thermochromism
Working mechanism		
Multicolor patterns	×	√
Pressure, temperature, and time for patterning	~4 MPa	3-12.3 kPa
	$T > T_g$	$T > T_m$
	> 35 min	< 1 min
Temperature and time for erasing	120 °C	80 °C
	10 min	1.8 s
Selective erasure	N.A	√
Resolution	N.A	7.2 μm
Rewriteability	√	√

*N. A. is not available.

Reference

1. Ji C, Chen M, Wu L. Patternable and Rewritable Retroreflective Structural Color Shape Memory Polymers. *Adv Optical Mater* **9**, 2100739 (2021).

Reference 1 reported a kind of shape memory retroreflection structural color film (SMRSCF) with a micrometer-sized (5-20 μm) dome monolayer array structures and showed its applications in generating patterns with the on-off color contrast. The SMRSCF exhibits interesting retroreflective structural color based on the combined effect of total internal reflection (TIR) and interference. Nevertheless, this specific color generation mechanism leads to unsatisfied optical performances: 1) more than one reflection peaks and thus poor color purity, and 2) no obvious regularity between the color and the size of the dome. The SMRSCF exhibits colorless recordability-erasability based on the SMRSCF's shape memory effect. The colorless recordability involves color disappearance along with the collapse of dome structures by pressing the SMRSCF above the glass transition temperature (T_g : 63 °C) and fixing the colorless state by cooling down. Then, the recorded colorless state can return to the pristine state after being heated

at 120 °C for 10 min. Reference 1 exhibits significant novelty within the areas of focus on the on-off retroreflective colors based on shape memory micro-domes. However, challenges, including impure colors, the difficulty in printing multicolor patterns, the requirements of high pressure (≈ 4 MPa), and the long time for colorless recording (35 min)-erasing (10 min) may limit SMRSCFs' practical applications.

In contrast, our work developed a new type of thermal-responsive mechanochromic photonic crystal (TRMPC) by combining phase-change materials (PCM) with a high melting point (T_m : 46 °C) and mechanochromic photonic crystals (MPCs) with non-closely packing structures and demonstrated its applications as ink-free rewritable paper for printing multicolor patterns. Owing to the efficient light diffraction by the non-close-packing structure with a small refractive index contrast, the TRMPC exhibits outstanding color purity and a broad tuning range of structural colors, favorable for practical applications. More importantly, the TRMPC shows unique multicolor recordability-erasability based on the unique thermoswitchable on-off mechanochromism. The color recordability is accomplished in two processes: color generation by pressing the TRMPC at $T > T_m$ and subsequently color fixation through a cooling down process ($T < T_m$). Multicolor can be recorded by adjusting pressure with similar procedures. The recorded color can be erased by heating the TRMPC at 80 °C in seconds. Different from SMRSCFs, TRMPCs possess unique optical performances: 1) TRMPCs can record multicolors (440-632 nm), in contrast to recording the colorless state of the SMRSCF; 2) only a low pressure (3-13.9 kPa) is required for multicolor recording, 287-1333 times smaller than that of the SMRSCF; 3) it only takes 2 s for the TRMPC to finish the color variation and color erasing, 1050 and 300 times quicker than those of the SMRSCF, respectively; and 4) multicolor and high-resolution patterns (line: 7.2 μm and dot: 5.4 μm) can be repeatedly printed on the TRMPC paper.

The above comparison between Ref 1 and this work firmly proves that these two works are quite different in material fabrication, structure design, color-generating mechanism, optical properties, color recordability-erasability, and printing patterns.

The above discussion has been supplied as Supplementary Note 3 in the supporting information, due to the limited number of words allowed in the manuscript. The related revision can be found in lines 302-305 and Ref 1 is cited as 32 (lines 477-478) in the revised manuscript.

(2) To efficiently achieve multicolor recording in a single operation, managing strain instead of applying pressure is considered to be more effective. Accordingly, it would be greatly appreciated if a strain-stress curve could be provided, illustrating how strain varies in response to stress. In this regard, providing a strain-stress curve that shows the change in strain according to stress would be very useful.

Author reply: We are grateful for your kind suggestion. Multicolor recordability also can be realized by stretching the TRMPC at $T > T_m$, followed by cooling down till $T < T_m$. When the stretching strain increases from 0 to 51.2%, the structural color of the TRMPC gel turns from red to blue (Fig. R10a) and the corresponding λ blueshifts from 631 to 441 nm (Fig. R10b-c). The tuning range of the wavelength ($\Delta\lambda$) by stretching is 190 nm, comparable to that (192 nm) by pressing. The sensitivity is 3.71 nm/%, much higher than those (1.11-3.16 nm/%) of highly sensitive MPCs. After cooling down, these colors can be recorded, as confirmed by their microscope images and λ . As expected, the recorded color can be erased by heating the TRMPC solid to $T > T_m$. These results proved that multicolor recordability can be realized by stretching during the color change process.

Fig. R10 Multicolor recordability based on stretching the TRMPC. **a** The microscope images of TRMPC gel and solid under different strain. **b** The reflection spectra of TRMPC gel under different strain. **c** The λ of TRMPC gel and solid as a function of strain.

The related revision can be found in lines 293-301 in the revised manuscript.

The strain-stress curves of TRMPC gel and solid are shown in Fig. R11. The TRMPC gel is elastic and easily stretchable under low stress, showing a maximal stretching strain of 58% corresponding to a stress of 51.4 kPa. In dramatic contrast, during the stretching process, the TRMPC solid first experiences 1) reversibly elastic deformation (strain: 0-12%, maximal stress: 847.7 kPa), and 2) then irreversibly plastic deformations (strain: 12-118%, stress: 792-848 kPa). The Young's modulus of the TRMPC solid is 22.86 MPa, nearly 260 times higher than that of the TRMPC gel, firmly demonstrating that the deformability of the non-close-packing structure is greatly limited by the solid PCM at room temperature.

Fig. R11 Fabrication and characterization of the TRMPC. g The stress-strain curves of the TRMPC solid and gel.

The related revision can be found in lines 195-201 in the revised manuscript.

(3) For the practical application of this inkless rewritable paper, resolution is an important factor. It is imperative to ascertain the minimal pressure necessary for effective patterning, as well as to delineate the achievable resolution of these patterns. This understanding is crucial in evaluating the paper's practical utility and functional performance.

Author reply: Thank you very much for your insightful comments. To ascertain the minimal pressure for effective patterns, the heart-like patterns with λ located at 615 and 600 nm were fabricated by applying respective 2.2 and 2.6 kPa (Fig. R12) on the typical red TRMPC (632 nm) during the color change process. As expected, these solid patterns show exceptional stability under normal conditions. The 2.6 kPa-based pattern can be easily recognized by its distinct color contrast,

while it is not easy to distinguish the 2.2 kPa-based pattern due to its small color contrast. Therefore, the minimum pressure for generating an effective pattern should be around 2.2 kPa.

Fig. R12 The minimal pressure for effective patterns. The λ of patterns as a function of time. The insets are the digital photos of the patterns.

We have tried to fabricate high-resolution patterns using elaborate stamps. Fig. R13 shows that the line and point resolutions that we can obtain are measured to be 7.2 and 5.4 μm , respectively. We believe that better resolution can be realized with more elaborate stamps.

Fig. R13 The maximum resolution of the patterns. a-c Schematic illustration of the preparation of micropatterns and the corresponding microscope images of micropatterns.

The related revision can be found in lines 327-330 in the revised manuscript.

REVIEWERS' COMMENTS

Reviewer #1 (Remarks to the Author):

Responses by the authors are appropriate. The authors fully responded to my previous concern and the quality of the manuscript is clearly improved. I would like to recommend publication of this manuscript in Nature Communications.

Reviewer #2 (Remarks to the Author):

The authors have effectively addressed my initial comments. However, I remain concerned about the advancement of this work relative to photonic crystals made from shape-memory polymers. In the revision, the comparison was made with a single reference, which I mentioned, concerning color patterning with micro-dome arrays utilizing TIR-based structural coloration. For a more comprehensive evaluation, it would be beneficial to include additional comparisons with other relevant works on colloidal photonic crystals made of shape-memory polymers, such as Nanoscale, 2019, 11, pp. 20015–20023.

Reviewer #3 (Remarks to the Author):

Response to reviewers

Reviewers 1 and 3 have co-reviewed this work: *Responses by the authors are appropriate. The authors fully responded to my previous concern and the quality of the manuscript is clearly improved. I would like to recommend the publication of this manuscript in Nature Communications.*

Author reply: We are very grateful to the reviewers for your kind suggestions on this work, which has greatly improved the quality of our manuscript. We also appreciate the reviewers' high praise and positive summary of this work.

Reviewer #2: *The authors have effectively addressed my initial comments. However, I remain concerned about the advancement of this work relative to photonic crystals made from shape-memory polymers. In the revision, the comparison was made with a single reference, which I mentioned, concerning color patterning with micro-dome arrays utilizing TIR-based structural coloration. For a more comprehensive evaluation, it would be beneficial to include additional comparisons with other relevant works on colloidal photonic crystals made of shape-memory polymers, such as *Nanoscale*, 2019, 11, pp. 20015-20023.*

Author reply: Thank you very much for your kind comments and suggestions. In the revised manuscript, we have compared our work with more references (1-5) in materials, structures, optical properties, and working mechanisms of color recordability and erasability, as listed in Table R1.

Table R1. Comparisons of materials, structures, and properties between references 1-5 and our work.

	Ref 1	Ref 2	Ref 3	Ref 4	Ref 5	Our work
Materials	Epoxy resin/PU/PA	PS@PEA@Pi BMA-co-PEA	ETPTA-co-PEGDA	PS-co-BA	SU-8	SiO ₂ /PEGPEA/DEG EEA/PCM
Structure	Closely arranged micro-dome monolayer	Close-packing opals	Inverse opal	Inverse opal	Inverse opal	Non-close-packing opals
Color-generating mechanism	TIR and interference	LD by opals	LD by inverse opals	LD by inverse opals	LD by inverse opals	LD by opals
Structural color	Retroreflection	Reflection	Reflection	Reflection	Reflection	Reflection
Color purity	Poor	Good	Good	Poor	Poor	Good
Working mechanism	Shape memory effect	Shape memory effect	Shape memory effect	Shape memory effect	Elastoplastic deformation	Thermoswitchable on-off mechanochromism
Tuning range of λ	×	120 nm	×	143 nm	165 nm	192 nm
Multicolor	×	N.A	×	√	√	√
Patterning requirements	~4 MPa $T > T_g$ > 35 min	5-50 % $T > T_g$ N.A	4.21-54.4 kPa 23 °C N.A	~18 MPa $T > T_g$ 2-10 s	17.6-20.4 MPa 25 °C 0.3 s	3-12.3 kPa $T > T_m$ < 1 min
Erasing requirements	120 °C 10 min	53 °C ~ 50 s	23 °C N.A	80 °C 8 min	25 °C ×	80 °C 1.8 s
Selective erasure	N.A	N.A	N.A	N.A	×	√
Resolution	N.A	N.A	N.A	N.A	N.A	7.2 μ m
Rewriteability	√	N.A	√	√	×	√

*N. A. is not available. TIR and LD represent total internal reflection and light diffraction, respectively.

References

1. Ji, C., Chen, M. & Wu, L. Patternable and rewritable retroreflective structural color shape memory polymers. *Adv Optical Mater* **9**, 2100739 (2021).
2. Wu, P., Shen, X., Schafer, C. G., Pan, J., Guo, J. & Wang, C. Mechanochromic and thermo chromic shape memory photonic crystal films based on core/shell nanoparticles for smart monitoring. *Nanoscale* **11**, 20015-20023 (2019).
3. Fang, Y., Ni, Y., Leo, S. Y., Taylor, C., Basile, V. & Jiang, P. Reconfigurable photonic crystals enabled by pressure-responsive shape-memory polymers. *Nat. Commun.* **6**, 7416 (2015).
4. Wang, Y., Zhao, Q. & Du, X. Inkless multi-color writing and copying of laser-programmable photonic crystals. *Mater. Horiz.* **7**, 1341-1347 (2020).
5. Cho, Y. et al. Elastoplastic inverse opals as power-free mechano chromic sensors for force recording. *Adv. Funct. Mater.* **25**, 6041-6049 (2015).

Reference 1 reported a shape memory retroreflection structural color film (SMRSCF) with a micrometer-sized (5-20 μm) dome monolayer array structures and showed its applications in generating patterns with the on-off color contrast. The SMRSCF exhibits interesting retroreflective structural color based on the combined effect of total internal reflection (TIR) and interference. Nevertheless, this specific color generation mechanism leads to unsatisfied optical performances: 1) more than one reflection peaks and thus poor color purity, and 2) no obvious regularity between the color and the size of the dome. The SMRSCF exhibits colorless recordability-erasability based on the SMRSCF's shape memory effect. The colorless recordability involves color disappearance along with the collapse of dome structures by pressing the SMRSCF above the glass transition temperature (T_g : 63 $^{\circ}\text{C}$) and fixing the colorless state by cooling down. Then, the recorded colorless state can return to the pristine state after being heated at 120 $^{\circ}\text{C}$ for 10 min. Reference 1 exhibits significant novelty within the areas of focus on the on-off retroreflective colors based on shape memory micro-domes. However, challenges, including impure colors, the difficulty in printing multicolor patterns, the requirements of high pressure (≈ 4 MPa), and the long time for colorless recording (35 min)-erasing (10 min) may limit SMRSCFs' practical applications.

Reference 2 developed core-shell polymer particles-based shape memory photonic crystals (SMPCs) with close-packing structures as smart labels for environmental monitoring, which were

fabricated by a facile hot-pressing method and post-photocuring technology. These SMPCs show mechanochromic and thermochromic properties owing to the soft shape memory shell of the polymer particle. Color recordability was realized by stretching the SMPC above the T_g (~ 53 °C) to adjust colors and subsequently fixing changed colors by cooling the stretched SMPC below the T_g . The fixed color could afterward be recovered by heating the stretched SMPC above the T_g for ~ 50 s. This work provides an interesting idea for preparing SMPCs by adjusting the properties of core-shell particles. Nevertheless, the SMPC suffers from a small tuning range of λ ($\Delta\lambda = 120$ nm), limited sensitivity (2.4 nm/%), low recovery time (~ 50 s), and a dramatic decrease in color saturation during stretching, restricting their practical applications as multicolor rewritable papers.

Reference 3 reported a stimuli-responsive shape-memory polymer (SMP) with inverse opal structures, which can realize cold programming and instantaneous shape recovery at the nanoscale. The pristine SMP exhibits brilliant color due to the ordered packing of micropores and large refractive index contrast between air and polymers. The macropores of the SMP collapse once compressing the SMP, leading to the disappearance of structural color and a transparent appearance. This specific working mechanism means that patterns with only a monochrome color can be recorded on the SMP paper. The recorded color can revert to the pristine state by immersing compressed SMPs in solvents with low surface tension (ethanol and toluene). Unfortunately, the monochrome color recordability and the use of organic vapors may limit their large-area applications.

Reference 4 proved the fabrication of a rewritable PC paper based on programmable configurations of a synthesized shape memory polymer (SMP). The SMP-based PC paper possesses compression-induced collapsed micropores, thus showing a colorless state. The collapsed micropores can be partially or fully recovered by controlling the time of laser-induced heating, which results in different colors. Multicolor patterns can be generated and recorded by region-selectively heating the PC papers to different temperatures using lasers and integrating a pre-printed photothermal layer. The recorded color can be erased by simply compressing the whole PC paper. This work offers a new possibility to record different colors by controlling the recovery degree of the PC's micropores. However, the unwanted diffusion colors caused by the inevitable conduction of heat at boundaries will restrict the color purity and the resolution of patterns.

Reference 5 reported elastoplastic inverse opals with the polymer SU-8 fabricated by infiltrating the colloidal crystals of silica particles with uncrosslinked SU-8 followed by the removal of the colloidal templates. The SU-8 films show a mechanochromic response that can change and record the reflective color depending on the magnitude and rate of the applied force. Due to the elastoplastic deformation of the SU-8 films, the deformed structures and thus colors can be locked after removing the load (17.6-20.4 MPa). Nevertheless, massive cracks on the SU-8 films lead to poor color purity. Moreover, the recorded colors cannot be erased due to the permanent structure change, which is unfavorable for rewritable materials.

Therefore, references 1-4 majorly reported the SMP-based photonic materials with closely arranged micro-dome monolayer, closely packed opal structures, and inverse opal structures. The keys to realizing color recordability and erasability are based on the intrinsic shape memory effect. In contrast, reference 5 reported the specific SU-8 with inverse opal structures capable of recording colors but not available for erasing colors, owing to the irreversible elastoplastic deformation of the SU-8 films.

In contrast, our work developed a type of thermal-responsive mechanochromic photonic crystal (TRMPC) by combining phase-change materials (PCM) with a high melting point (T_m : 46 °C) and mechanochromic photonic crystals (MPCs) with non-closely packing structures and demonstrated its applications as ink-free rewritable paper for printing multicolor patterns. Owing to the efficient light diffraction by the non-close-packing structure with a small refractive index contrast, the TRMPC exhibits outstanding color purity and a broad tuning range of structural colors, favorable for practical applications. More importantly, the TRMPC shows multicolor recordability-erasability based on the unique thermoswitchable on-off mechanochromism. The color recordability is accomplished in two processes: color generation by pressing the TRMPC at $T > T_m$ and subsequently color fixation through a cooling down process ($T < T_m$). Multicolor can be recorded by adjusting pressure with similar procedures. The recorded color can be erased by heating the TRMPC at 80 °C in seconds. High-resolution patterns (line: 7.2 μm and dot: 5.4 μm) can be repeatedly printed on the TRMPC paper.

The above discussion proves the significant differences between our work and references 1-5 in material fabrication, structure design, color-generating mechanism, optical properties, color recordability-erasability, and printing patterns.

The above discussion has been supplied as Supplementary Note 3 in the supporting information, due to the limited number of words allowed in the manuscript. The related revision can be found in lines 65-68, 305-307, and Ref 1-2 are cited as 32 and 33 (lines 479-483) in the revised manuscript.